# SplitFlow: Flow Decomposition for Inversion-Free Text-to-Image Editing

**Sung-Hoon Yoon[1]***, **Minghan Li[1]***, **Gaspard Beaudouin[2]**, **Congcong Wen[1,3]**,
**Muhammad Rafay Azhar[1]**, and **Mengyu Wang[1,4]†**

[1]Harvard AI and Robotics Lab, Harvard University
[2]École des Ponts, Institut Polytechnique de Paris [3]New York University Abu Dhabi
[4]Kempner Institute for the Study of Natural and Artificial Intelligence, Harvard University
{syoon13,mli4}@meei.harvard.edu, gaspard.beaudouin@eleves.enpc.fr,
cwen2@meei.harvard.edu, rafayazhar@college.harvard.edu,
mengyu_wang@meei.harvard.edu

## Abstract

Rectified flow models have become a *de facto* standard in image generation due to their stable sampling trajectories and high-fidelity outputs. Despite their strong generative capabilities, they face critical limitations in image editing tasks: inaccurate inversion processes for mapping real images back into the latent space, and gradient entanglement issues during editing often result in outputs that do not faithfully reflect the target prompt. Recent efforts have attempted to directly map source and target distributions via ODE-based approaches without inversion; however, these methods still yield suboptimal editing quality. In this work, we propose a flow decomposition-and-aggregation framework built upon an inversion-free formulation to address these limitations. Specifically, we semantically decompose the target prompt into multiple sub-prompts, compute an independent flow for each, and aggregate them to form a unified editing trajectory. While we empirically observe that decomposing the original flow enhances diversity in the target space, generating semantically aligned outputs still requires consistent guidance toward the full target prompt. To this end, we design a projection and soft-aggregation mechanism for flow, inspired by gradient conflict resolution in multi-task learning. This approach adaptively weights the sub-target velocity fields, suppressing semantic redundancy while emphasizing distinct directions, thereby preserving both diversity and consistency in the final edited output. Experimental results demonstrate that our method outperforms existing zero-shot editing approaches in terms of semantic fidelity and attribute disentanglement. The code is available at https://github.com/Harvard-AI-and-Robotics-Lab/SplitFlow.

## 1 Introduction

Flow-based generative models have demonstrated superiority in synthesizing images and also in text-to-image generation task. Building on these advances, recent research has actively explored image editing methods that modify a given image to align with a target prompt. Due to the nature of diffusion [20, 7, 23] and flow-based generative models [1, 13, 15], which generate samples from noise through iterative refinement, editing typically requires an inversion step to estimate the corresponding noisy latent representation. However, this inversion process is often imperfect and fails to recover an exact latent that reconstructs the original image. As a result, the editing process may suffer from semantic drift, visual distortion, or inconsistent attribute manipulation. Even

---

* Equal contribution. † Corresponding author.

39th Conference on Neural Information Processing Systems (NeurIPS 2025).

when a precise inversion allows near-perfect reconstruction, an empirical trade-off arises between reconstructability and editability. Latents that are highly optimized to match the input image tend to be rigid and entangled, meaning that adjusting a single attribute often leads to unintended changes in unrelated features [2].

Recently, some efforts have been made to improve the editing process based on rectified flow models [19, 26, 3, 29, 31] and also inversion-free method that directly maps the source and target distribution [11], but these methods show limited semantic fidelity due to gradient entanglement and prompt-latent misalignment. In particular, when a complex target prompt contains multiple semantic attributes (e.g., "a german shepard dog with black sunglasses jumping on the grass with mouth opened"), a single editing trajectory guided by the full prompt often leads to entangled gradients and conflicting directions in the semantic space. This makes it difficult to isolate and control the influence of individual attributes, resulting in either under-edited or overly distorted outputs.

Motivated by these issues, we propose **SplitFlow**, which decomposes the flow induced by the target prompt into independent sub-target flows derived from semantically decomposed sub-prompts. By computing independent editing flows for each sub-prompt and aggregating them into a unified trajectory while mitigating flow conflict, our method achieves improvements in both fidelity and editability. Performance on the PIE-Bench benchmark [10] demonstrates that the proposed approach outperforms prior methods.

The contributions of this paper are threefold:

- We demonstrate that decomposing the editing flow improves the fidelity of the edited image, especially in preserving background consistency.
- We propose SplitFlow, which progressively approximates the target latent through flow decomposition followed by flow aggregation.
- We introduce a projection-and-aggregation method that aligns sub-target flows with the target direction, while preserving their semantic diversity and alleviating potential conflicts velocity field that arise during integration.

## 2   Related Work

**Diffusion and Rectified Flow.**   Diffusion models [21, 18] synthesize images by gradually reversing a noising process that transforms data into Gaussian noise. Starting from random noise, they iteratively denoise through learned score functions, generating high-quality outputs over many timesteps. Rectified Flow (RF) [13, 15] introduces a more stable and efficient generation process via a straight trajectory in latent space.

**Text-to-Image Inversion and Editing.**   Text-guided image editing aims to modify a given image to match a target text prompt. Most diffusion-based editing pipelines rely on first mapping the image back into the model's latent space, typically in the form of Gaussian noise. DDIM [21] enables approximate inversion via a deterministic trajectory, but suffers from cumulative errors due to its linear approximation. To improve inversion accuracy, optimization-based approaches [16, 25, 8] have been proposed, though at a high computational cost. On the editing side, early methods achieve localized edits by fine-tuning diffusion models [5, 12] or manipulating cross-attention maps [6, 24]. While effective, these methods still operate within the diffusion framework and depend heavily on either inversion or fine-tuning, which are often inefficient for practical use.

Recent efforts have explored RF models for editing, which offer more stable and efficient sampling trajectories than diffusion models. RF-Inversion [19] formulates editing as an optimal control problem to balance editability and fidelity, RF-Solver [26] incorporates attention injection to guide editing, and FireFlow [3] employs higher-order ODE solvers to improve inversion accuracy. Despite their advantages, these methods still depend on iterative re-noising procedures, making them prone to cumulative error. FTEdit [29] reduces errors by performing iterative average at each inversion timestep, which is basically equivalent to increasing the number of sampling steps.

**Inversion-Free Editing.**   Unlike traditional inversion-based approaches, which require recovering the noise latent that originally generated the image, inversion-free methods bypass this step and directly optimize in the image or latent space. InfEdit [30] introduces an early inversion-free

framework based on the consistency models [22]. FlowEdit [11] extends this idea to RF models by directly manipulating image-space velocity differences. However, it lacks directional selectivity and often struggles with global edits. These works highlight that explicitly modeling semantic flow differences enhances editing control and efficiency.

**Multi-Task and Flow Decomposition.** Our method builds upon insights from gradient conflict resolution in multi-task learning [32, 17, 14], where conflicting objectives are resolved via adaptive re-weighting. Similarly, we propose to semantically decompose the target prompt into multiple sub-flows and adaptively aggregate them for consistent and diverse edits. To our knowledge, our work is the first to incorporate flow decomposition and aggregation into text-based image editing framework.

# 3 Preliminaries

## 3.1 Flow Matching and Rectified Flow

Let $\mathbb{R}^d$ denote the data space, and let $x_0 \in \mathbb{R}^d$ be an initial data point sampled from a source distribution $p_0$. Flow Matching (FM) methods [15, 13, 1] aim to learn a time-dependent vector field $v_\theta(x_t, t) : [0, 1] \times \mathbb{R}^d \to \mathbb{R}^d$, such that the solution to the following ODE transports $x_0 \sim p_0$ to $x_1 \sim p_1$, where $p_1$ denotes the target distribution:

$$dx_t = v_\theta(x_t, t)dt \tag{1}$$

The solution $x_t$ describes a continuous trajectory defined by the ODE, starting from the initial point $x_0$ and reaching the target point $x_1$. The ground-truth vector field $v^*(x_t, t)$ governs this trajectory and induces a distributional path $x_t$ that satisfies the boundary conditions $x_{t=0} = x_0$ and $x_{t=1} = x_1$. The objective of FM is to learn a parameterized vector field $v_\theta(x_t, t)$ that approximates $v^*(x_t, t)$ by minimizing the following regression loss:

$$\mathcal{L}(\theta) = \mathbb{E}_{x_t, t}[\|v_\theta(x_t, t) - v^*(x_t, t)\|_2^2], \tag{2}$$

where $\|\cdot\|_2^2$ denotes mean square error. Here, FM enables deterministic sampling via ODEs, avoiding the stochastic noise accumulation seen in diffusion models. By leveraging known couplings between distributions, it ensures both interpretability and fidelity. This leads to faster, more stable sampling and a principled connection to optimal transport, making FM a strong candidate for generative modeling. Rectified Flow (RF) [15], a special case of FM, aims to learn an ODE whose solution closely follows straight-line trajectories between pairs of points sampled from $x_0$ and $x_1$. The ground-truth vector field is defined as $v^*(x_t, t) = x_1 - x_0$, where $x_t$ is the linear interpolation between $x_0$ and $x_1$. The training objective can thus be formulated as:

$$\mathcal{L}(\theta) = \mathbb{E}_{x_t, t} \|v_\theta(x_t, t) - (x_1 - x_0)\|_2^2, \quad \text{where} \quad x_t = (1 - t)x_0 + tx_1. \tag{3}$$

By eliminating the need for stochastic sampling and instead following deterministic linear paths, RF enables faster and more stable performance in downstream tasks such as ODE-based generative modeling and image editing.

## 3.2 Inversion-free Image Editing

Image editing aims to transform a source image $x_0^{src}$, guided by a source text prompt $\varphi^{src}$, into an edited image $x_0^{tgt}$ using a target text prompt $\varphi^{tgt}$. In RF, the trajectory between noise and image is assumed to be linear, meaning that the noisy latent at any timestep can be obtained via linear interpolation. Leveraging this property, inversion-free image editing method [11] bypass the need for latent inversion by estimating the vector field difference between the source and target images at each timestep. This enables the construction of a transition path in the clean image space, gradually mapping the source image toward the target. Specifically, the target-aligned latent at timestep $t$, denoted as $x_t^{\mathrm{FE}}$, can be approximated by:

$$x_t^{\mathrm{FE}} = x_0^{src} + x_t^{tgt} - x_t^{src}. \tag{4}$$

The evolution of this trajectory follows the ODE:

$$dx_t^{\mathrm{FE}} = v_\theta^{\Delta}(x_t^{tgt}, x_t^{src}) \, dt \approx v_\theta^{\Delta}(x_{t-1}^{\mathrm{FE}} + x_t^{src} - x_0^{src}, x_t^{src}) \, dt, \tag{5}$$

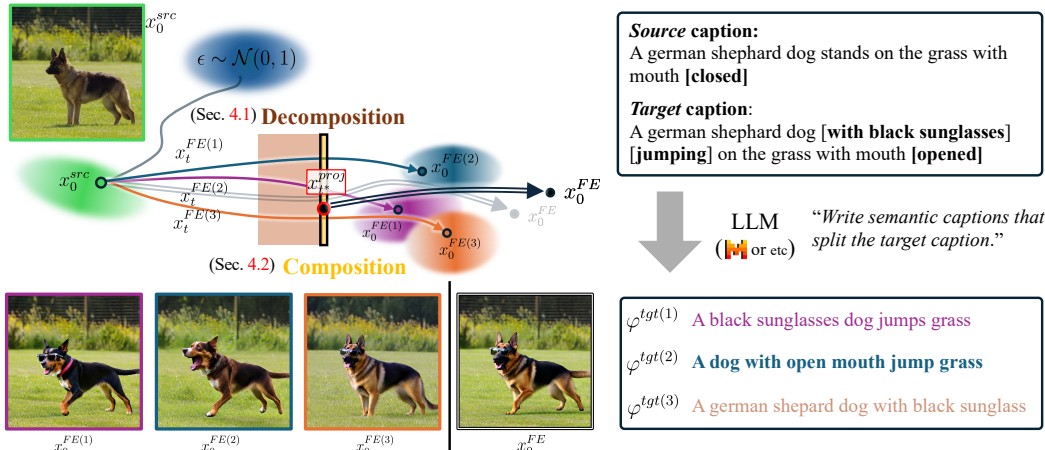

Figure 1: Simplified visual illustration of the proposed SplitFlow. *Purple, Blue, Orange* line indicates independent sub-target flow. The aggregation is done on a certain timestep. After the aggregation, we use a single, unified flow. The notation in this figure follows the paper.

where the difference in velocity fields guided by the source and target prompts is defined as: $v_t^\Delta(x_t^{tgt}, x_t^{src}) = v_\theta(x_t^{tgt}, t, \varphi^{tgt}) - v_\theta(x_t^{src}, t, \varphi^{src})$. Since the source image is known, $x_t^{src}$ can be directly obtained by linearly interpolating between it and a randomly sampled noise. In contrast, the target image is unknown and cannot be directly interpolated. Therefore, $x_t^{tgt}$ is approximated using the previous timestep $x_{t-1}^{FE}$ and Eq. (4) as follows: $x_t^{tgt} \approx x_{t-1}^{FE} + x_t^{src} - x_0^{src}$.

Based on the estimated velocity field difference, the entire editing process can be implemented as an iterative trajectory in the clean image space. Starting from the source image, we initialize the path with $x_0^{FE} = x_0^{src}$. At each timestep $t$, the edited latent is updated using the approximated velocity difference: $x_{t-1}^{FE} = x_t^{FE} + \Delta_t v_\theta^\Delta(x_t^{tgt}, x_t^{src})$. This iterative procedure continues until the trajectory converges to the desired target image.

# 4 Method

Long target prompts often contain multiple attributes and complex semantics, resulting in a large semantic gap between the source and target in the latent space. This gap makes direct editing challenging, as entangled gradients can degrade edit quality. Providing simultaneous guidance for all attributes may lead to *conflicting flows*, which can cause semantic drift or even failure in the editing process. To address this challenge, we propose **SplitFlow**, an editing framework that progressively approximates the target latent through *flow decomposition* (Sec. 4.1) followed by *flow composition* (Sec. 4.2). Specifically, we first decompose the semantic complexity of the target prompt into a set of sub-target prompts and compute an independent flow for each, enabling latent directional components to be isolated and manipulated separately. These sub-flows are then aggregated into a unified flow that semantically aligns with the original target prompt. The editing process proceeds along this unified trajectory, resulting in more stable, diverse edits.

## 4.1 Progressive Target Approximation with Flow Decomposition

In image editing tasks, the semantic gap between a source prompt $\varphi^{src}$ and a target prompt $\varphi^{tgt}$ is often complex and high-dimensional. Direct transformation from source to target may lead to unstable or imprecise editing results. To address this, we propose to decompose the overall semantic transition into a sequence of intermediate sub-target prompts $\{\varphi^{tgt(i)}\}_{i=1}^N$, where $N$ is the number of sub-target prompts. This decomposition simplifies the editing task into semantically controllable and progressively guided transformations.

To perform the decomposition, we leverage a Large Language Model (LLM) as a prompt reasoning engine, represented as a function $f_{\text{LLM}}$. We construct a composite input sequence by concatenating an instruction prompt $\psi$, the source prompt $\varphi^{src}$, and the target prompt $\varphi^{tgt}$. Then we feed the input

to $f_{\text{LLM}}$ to generate a sequence of sub-target prompts that represent incremental semantic transitions:

$$\{\varphi^{tgt(i)}\}_{i=1}^{N} = f_{\text{LLM}}(\Phi), \quad \Phi = \text{concat}[\psi, \varphi^{src}, \varphi^{tgt}]. \tag{6}$$

Each sub-target prompt $\varphi^{tgt(i)}$ captures a localized semantic component that contributes to the overall transformation from $\varphi^{src}$ to $\varphi^{tgt}$. For example, as shown in Fig. 1, given a source prompt $\varphi^{src} =$ *"A german shepherd dog stands on the grass with mouth **closed**"* and a target prompt $\varphi^{tgt} =$ *"A german shepherd dog with black sunglasses jumping on the grass with mouth opened"*, and instruction prompt $\psi =$ *"Write semantic captions that split the target caption."*, the resulting sub-target prompts may be: $\{\varphi^{tgt(i)}\}_{i=1}^{N} = \{$ "A black sunglasses dog jumps grass", "A dog with open mouth jump grass", "A german shepherd with black sunglasses" $\}$. The number of sub-target prompts $N$ is adaptively determined by $f_{\text{LLM}}$ based on the degree and complexity of the semantic difference. In most cases, $N \leq 3$, which yields a compact yet effective semantic trajectory for guided editing.

Following the formulation of the baseline (Eq. (4)–(5)), we extend the framework to handle each sub-target prompt individually. For each sub-target prompt $\varphi^{tgt(i)}$, we define a corresponding flow $x_t^{\text{FE}(i)}$ governed by an independent velocity field. Specifically, the sub-target flow is expressed as follows:

$$x_t^{\text{FE}(i)} = x_0^{src} + x_t^{tgt(i)} - x_t^{src}, \tag{7}$$

where $x_0^{src}$ denotes the initial latent of the source image, $x_t^{src}$ is the interpolated latents at timestep $t$ between the source image and a Gaussian noise. Accordingly, the ODE governing each decomposed sub-target flow is given by the sub-target relative velocity field $v_t^{\Delta}(x_t^{tgt(i)}, x_t^{src})$:

$$dx_t^{\text{FE}(i)} = v_t^{\Delta}(x_t^{tgt(i)}, x_t^{src}) \cdot dt, \quad v_t^{\Delta}(x_t^{tgt(i)}, x_t^{src}) = v_\theta(x_t^{tgt(i)}, t, \varphi^{tgt(i)}) - v_\theta(x_t^{src}, t, \varphi^{src}).$$

Since $x_t^{tgt(i)}$ is not directly observable during inference, we approximate it based on the previously updated latent in Eq. (7) as: $x_t^{tgt(i)} \approx x_{t-1}^{\text{FE}(i)} + x_t^{src} - x_0^{src}$. The decomposition phase starts from $\eta_{max}$ and proceeds until $\eta_{dec}$.

## 4.2 Flow Composition

While decomposing the flow, as described in Sec. 4.1, allows us to disentangle the gradients and achieve independent transformations, the ultimate objective in the image editing task is to perform editing aligned with the target prompt faithfully. Therefore, we devise a method to compose the previously generated sub-target flows into a unified flow that adheres to the full semantics of the target prompt. Considering the formulation of the ODE as shown in Eq. 1, the velocity field computed for each sub-target prompt can be interpreted as a gradient-like vector in latent space that guides the latent representation toward the target state. In multi-task learning (MTL), it is well known that gradients from different tasks can conflict, leading to unstable optimization or degraded performance on each tasks. To address this issue, various strategies such as gradient projection and orthogonality constraints have been proposed. Inspired by gradient conflict resolution methods [17, 14] in MTL, we propose a method that mitigates interference between sub-target velocity fields while effectively achieving the desired image editing objectives.

### 4.2.1 Latent Trajectory Projection (LTP).

To enforce global semantic consistency with the target flow while maintaining the diversity of each sub-target flow, we project each sub-target latent (conditioned on $\varphi^{tgt(i)}$) onto the target latent (conditioned on $\varphi^{tgt}$): $x_t^{\text{FE}}$. To perform this projection, we normalize the target latent as $\hat{x}_t^{\text{FE}} = x_t^{\text{FE}}/\|x_t^{\text{FE}}\|_2$. Here, the projection of each sub-target latent $\{x_t^{\text{FE}(i)}\}_{i=1}^{N}$ onto $\hat{x}_t^{\text{FE}}$ is computed as follows:

$$x_t^{\text{proj}(i)} = \left( \langle x_t^{\text{FE}(i)}, \hat{x}_t^{\text{FE}} \rangle \right) \hat{x}_t^{\text{FE}}, \tag{8}$$

where the inner product ($\langle \cdot, \cdot \rangle$) is computed along the channel dimension of the latent. By projecting the sub-target latent onto the target latent, we ensure that the overall editing process remains consistent with the intended semantic shift. We then aggregate the projected sub-target latent to form the *projected latent*:

$$x_t^{\text{proj}} = \frac{1}{N} \sum_{i=1}^{N} x_t^{\text{proj}(i)}. \tag{9}$$

While both the target latent $x_t^{tgt}$ and the projected sub-target latents $\{x_t^{\text{proj}(i)}\}_{i=1}^N$ are aligned along the same semantic direction, their origins differ fundamentally: the former stems from a unified trajectory conditioned on the full target prompt, whereas the latter are partial trajectories derived from sub-prompts and aligned through projection. This approach retains the global coherence of the target trajectory while preserving localized variations introduced by sub-target prompts, enhancing both semantic consistency and editing diversity.

### 4.2.2 Velocity Field Aggregation (VFA).

After projection, to further enhance flow diversity, we introduce Velocity Field Aggregation (VFA), which combines the velocity fields of sub-target flows. To quantify the directional consistency among sub-target flows, we compute the cosine similarity between their relative velocity fields with respect to the source latent. Specifically, we first compute the relative velocity vector between $x_t^{\text{proj}(i)}$ and $x_t^{src}$ as follows:

$$\mathbf{g}_i := v_t^{\Delta}(x_t^{\text{proj}(i)}, x_t^{src}) = v_\theta(x_t^{\text{proj}(i)}, t, \varphi^{tgt(i)}) - v_\theta(x_t^{src}, t, \varphi^{src}) \tag{10}$$

The cosine similarity $\mathbf{S}_{ij}$ between the $i$-th and $j$-th sub-target prompt is defined as:

$$\mathbf{S}_{ij} = \left\langle \hat{v}_t^{\Delta(i)}, \hat{v}_t^{\Delta(j)} \right\rangle, \quad \hat{v}_t^{\Delta(i)} = \frac{v_t^{\Delta}(x_t^{\text{proj}(i)}, x_t^{src})}{\|v_t^{\Delta}(x_t^{\text{proj}(i)}, x_t^{src})\|}. \tag{11}$$

Here, $\hat{v}_t^{\Delta(i)}$ is the normalized from $v_t^{\Delta}(x_t^{proj(i)}, x_t^{src})$ to compute cosine similarity. This metric captures the angular agreement between projected velocity directions, thereby reflecting the consistency of semantic changes introduced by each sub-target prompt. By applying the softmax operation to the cosine similarity map $S \in \mathbb{R}^{N \times N \times H \times W}$, we obtain a weight map $w \in \mathbb{R}^{N \times H \times W}$ that determines the relative contribution of each velocity field at every spatial location $(h, w)$ in the latent grid:

$$\bar{v}_t^{\Delta}(h, w) = \sum_{i=1}^{N} w_i(h, w) \cdot v_t^{\Delta(i)}(h, w), \quad w_i(h, w) = \frac{\exp\left(\sum_{j \neq i} \mathbf{S}_{ij}(h, w)\right)}{\sum_k \exp\left(\sum_{j \neq k} \mathbf{S}_{kj}(h, w)\right)}. \tag{12}$$

Combining the projected latent from Eq. (9) with the aggregated velocity field $\bar{v}_\theta^{\Delta}$, the latent after aggregation can be updated as follows:

$$x_t^{\text{FE}} \leftarrow x_t^{\text{proj}} + \bar{v}_t^{\Delta} \cdot dt. \tag{13}$$

Since the proposed VFA adaptively weights the sub-target velocity fields based on their semantic alignment, it not only suppresses the influence of redundant flows but also emphasizes those with distinct semantic directions. This enables the model to preserve editing diversity while maintaining coherent alignment with the target prompt. Also, note that the LTP and VFA are applied at the end of the decomposition phase ($\eta_{dec}$).

**Mathematical Justification of VFA.** Here, we mathematically verify why VFA improves both fidelity and editability over mere averaging by showing:

$$\langle \bar{\mathbf{g}}, \mathbf{g}_{avg} \rangle \geq \|\mathbf{g}_{\text{avg}}\|^2, \tag{14}$$

where $\bar{\mathbf{g}} = \sum_{i=1}^{K} w_i \mathbf{g}_i$ and $\mathbf{g}_{avg} = \frac{1}{K} \sum_{i=1}^{K} \mathbf{g}_i$. Here, $S_{kj}$ is denoted as $\langle \hat{\mathbf{g}}_k, \hat{\mathbf{g}}_j \rangle$ and $a_k = \sum_j S_{kj}$. We first reformulate Eq. (14) in terms of the scores $a_i$, where the LHS is $\langle \bar{g}, g_{avg} \rangle = \frac{1}{K} \sum_{i=1}^{K} w_i a_i$ and the RHS is $\|g_{\text{avg}}\|^2 = \frac{1}{K^2} \sum_{i=1}^{K} a_i$. Thus, the proposition is equivalent to proving $\sum_i w_i a_i \geq \frac{1}{K} \sum_i a_i$. The proof combines two standard results. First, from Gibbs' inequality, the KL-divergence between the softmax distribution $w = \{w_i\}$ and the uniform distribution $u = \{1/K\}$ is non-negative, which implies: $\sum_{i=1}^{K} w_i a_i \geq \log(Z/K)$ (14-1), where $Z = \sum_i e^{a_i}$. Second, applying Jensen's inequality to the convex function $f(x) = e^x$ gives $\log(E[e^a]) \geq E[a]$, which in our context is:

$$\log(Z/K) = \log\left(\frac{1}{K} \sum_{i=1}^{K} e^{a_i}\right) \geq \frac{1}{K} \sum_{i=1}^{K} a_i \tag{14-2}$$

Table 1: Quantitative comparison results on PIE benchmark. For each model group (diffusion-based and flow-based), the best and second-best values are indicated in bold and underlined, respectively. Ours$^\dagger$ is the result with a fidelity-enhanced setting.

| Method | Model | Structure | | Background Preservation | | | | CLIP Similarity | |
|---|---|---|---|---|---|---|---|---|---|
| | | Editing | Distance $_{\times 10^3}$ ↓ | PSNR ↑ | LPIPS $_{\times 10^3}$ ↓ | MSE $_{\times 10^4}$ ↓ | SSIM $_{\times 10^2}$ ↑ | Whole ↑ | Edited ↑ |
| **DDIM** [21] | Diffusion | P2P | 69.4 | 17.87 | 208.80 | 219.88 | 71.14 | 25.01 | 22.44 |
| **DDIM** [21] | Diffusion | PnP | 28.22 | 22.28 | 113.46 | 83.64 | 79.05 | 25.41 | 22.55 |
| **Null-text** [16] | Diffusion | P2P | 13.44 | 27.03 | 60.67 | 35.86 | 84.11 | 24.75 | 21.86 |
| **PnP-Inv** [10] | Diffusion | P2P | **11.65** | **27.22** | **54.55** | **32.86** | **84.76** | 25.02 | 22.10 |
| **PnP-Inv** [10] | Diffusion | PnP | 24.29 | 22.46 | 106.06 | 80.45 | 79.68 | **25.41** | **22.62** |
| **RF-Inversion** [19] | Flux | - | 40.6 | 20.82 | 184.8 | 129.1 | 71.92 | 25.20 | 22.11 |
| **RF-Solver** [26] | Flux | RF-Solver | 31.1 | 22.90 | 135.81 | 80.11 | 81.90 | 26.00 | 22.88 |
| **FireFlow** [3] | Flux | RF-Solver | 28.3 | 23.28 | 120.82 | 70.39 | 82.82 | 25.98 | 22.94 |
| **Flow Edit** [11] | Flux | - | 27.7 | 21.91 | 111.70 | 94.0 | 83.39 | 25.61 | 22.70 |
| **FTEdit** [29] | SD3.5 | AdaLN | 18.17 | 26.62 | 80.55 | 40.24 | **91.50** | 25.74 | 22.27 |
| **iRFDS** [31] | SD3 | - | 62.72 | 19.61 | 186.39 | 179.76 | 74.59 | 24.54 | 21.67 |
| **FlowEdit** [11] | SD3 | - | 27.24 | 22.13 | 105.46 | 87.34 | 83.48 | 26.83 | 23.67 |
| **FlowEdit** [11] | SD3.5 | - | 11.80 | 26.97 | 53.68 | 31.23 | 89.70 | 26.18 | 22.88 |
| **SplitFlow(Ours)** | SD3 | - | 25.96 | 22.45 | 102.14 | 81.99 | 83.91 | **26.96** | **23.83** |
| **SplitFlow(Ours)**$^\dagger$ | SD3 | - | 14.55 | 25.22 | 68.53 | 44.96 | 87.54 | 26.23 | 23.01 |
| **SplitFlow(Ours)** | SD3.5 | - | **11.68** | **27.12** | **52.93** | **30.61** | 89.76 | 26.29 | 22.89 |

Chaining inequalities Eq. (14-1) and Eq. (14-2) directly yields the required result:

$$\sum_{i=1}^{K} w_i a_i \geq \log(Z/K) \geq \frac{1}{K} \sum_{i=1}^{K} a_i$$

This proof formalises why VFA improves both fidelity and editability: It suppresses conflicts, steering the update toward high-consensus attributes. Empirically, this manifests as higher background preservation and better CLIP similarity in Table 1-2.

When the decomposition phase is ended, the aggregated latent now follows the ODE formulation of the target prompt as described in Sec. 3.2. The decomposed flow stages facilitate fine-grained attribute manipulation without gradient entanglement, while the final unified flow phase ensures alignment with the holistic editing intent. This hybrid strategy improves editing stability and fidelity by integrating diversity-aware adjustments with prompt-level consistency. A detailed algorithmic description of SplitFlow is included in the supplementary material.

# 5 Experiments

## 5.1 Experimental Setup

**Implementation Details.** To show the effectiveness of the proposed method and for a fair comparison with the prior works, we employed the commonly used Stable Diffusion (SD3, SD3.5) [4] rectified flow model. By following the protocol of the baseline [11], we use the same T = 50 steps with $\eta_{max} = 33$, which skips the first one-third steps. The CFG values for the source and target are set to 3.5 and 13.5, respectively. The decomposition $\eta_{dec}$ is set to 28, which lasts for **5 steps** ($\eta_{max} - \eta_{dec}$). To decompose the target prompt, we used Mistral-7B [9].

**Dataset.** We evaluate our method on Prompt-based Image Editing (PIE) Benchmark [10], which contains 700 images of natural and artificial scenes. In PIE benchmark, ten categories span from random editing written by volunteers to change image style. Each image includes a source prompt, target prompt, editing directive, edit subjects, and editing mask.

**Baselines and Evaluation Metrics.** As this work focuses on image editing based on rectified flow, we compare with the State-of-The-Art (SoTA) editing methods; RF-Inversion [19], RF-Solver [26], FireFlow [3], iRFDS [31], FTEdit [29], and FlowEdit [11]. Diffusion-based models such as DDIM [21, 16, 10] are also included for comparison. We evaluate our method on the PIE-Bench dataset using standard metrics. For assessing reconstruction quality and background preservation, we report image-level metrics: LPIPS [33], SSIM [27], MSE, PSNR, and Structure Distance [10]. To measure semantic alignment with the target prompt, we use CLIP similarity [28].

Table 2: Ablation study on PIE-benchmark. Here *AVG* denotes the case where mere average is applied for latent trajectory aggregation.

| Baseline | LTP | VFA | Structure | Background Preservation | | | | CLIP Similarity | |
|---|---|---|---|---|---|---|---|---|---|
| | | | Distance $_{\times 10^3}$ ↓ | PSNR ↑ | LPIPS $_{\times 10^3}$ ↓ | MSE $_{\times 10^4}$ ↓ | SSIM$_{\times 10^2}$ ↑ | Whole ↑ | Edited ↑ |
| ✓ | | | 27.24 | 22.13 | 105.46 | 87.34 | 83.48 | 26.83 | 23.67 |
| *AVG* | ✓ | | 22.28 | 23.36 | 92.00 | 68.26 | 85.00 | 26.81 | 23.67 |
| ✓ | ✓ | | 26.22 | 22.37 | 103.58 | 83.67 | 83.76 | 26.93 | 23.82 |
| ✓ | ✓ | ✓ | 25.96 | 22.45 | 102.14 | 81.99 | 83.91 | 26.96 | 23.83 |

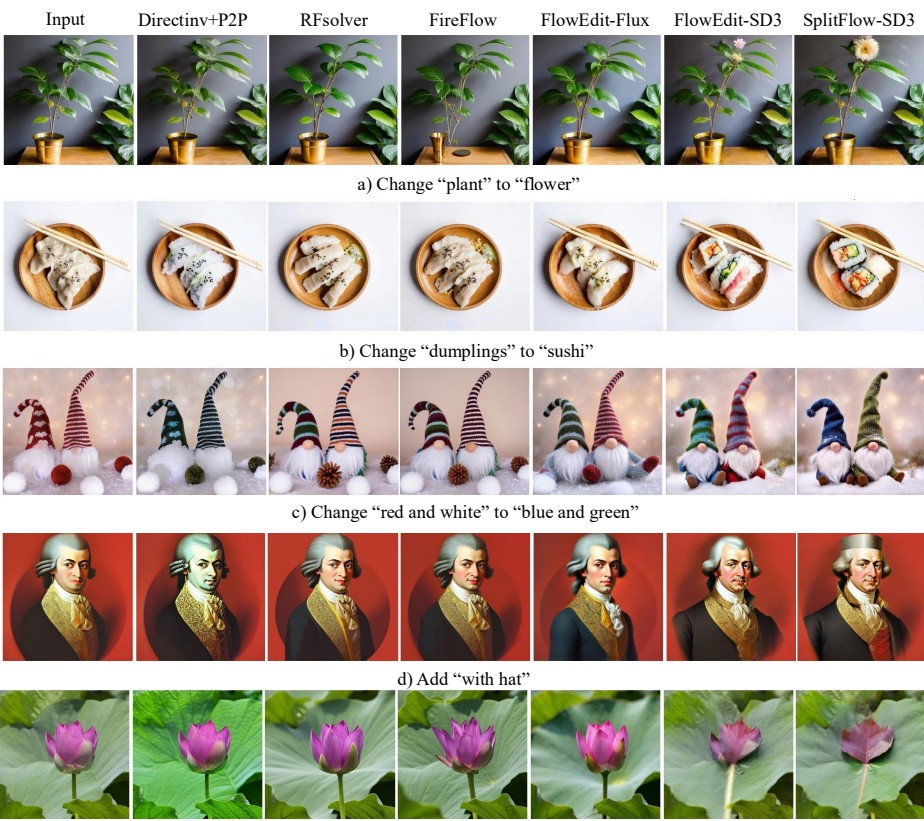

a) Change "plant" to "flower"

b) Change "dumplings" to "sushi"

c) Change "red and white" to "blue and green"

d) Add "with hat"

e) Delete "a single pink lotus flower is growing in the middle of "

Figure 2: Qualitative comparison of prompt-based image editing methods. Each row corresponds to a specific editing instruction, where the source prompt is modified into a target prompt. From top to bottom, the tasks are: (a) *change* "plant" to "flower", (b) *change* "dumpling" to "sushi", (c) *change* "red and white" to "blue and green", (d) *add* "with hat", (e) *delete* "a single pink lotus flower is growing in the middle of". The columns show the input image and the results generated by different models, including Directinv+P2P, RFsolver, FireFlow, FlowEdit-Flux, FlowEdit-SD3, and SplitFlow.

## 5.2 Main Results

**Comparison with State-of-the-art Methods.** To demonstrate the effectiveness of the proposed SplitFlow, we conducted experiments as shown in Table 1. Compared to the FlowEdit [11] and iRFDS [31], within the same SD3 model, the proposed SplitFlow not only outperforms in preserving background but also in editing quality. To further demonstrate the effect of the trade-off between fidelity and editability, we omit the noise interpolation during the decomposition phase and directly use the source latent $x_0^{src}$ to better preserve structural details of the input image in the fidelity-enhanced setting[†]. Compared to the state-of-the-art method FTEdit [29], our SplitFlow[†] (a fidelity-enhanced variant) achieves superior performance in both Structure Distance and LPIPS, despite FTEdit employing a stronger backbone model. Moreover, it significantly outperforms FTEdit in both CLIP Similarity metrics, demonstrating better alignment with the target prompt. Also, compared

Table 3: Ablation study of aggregation step $\eta_{dec}$ on PIE-benchmark.

| $\eta_{dec}$ | Structure | Background Preservation | | | | CLIP Similarity | |
|---|---|---|---|---|---|---|---|
| | Distance $_{\times 10^3}$ ↓ | PSNR ↑ | LPIPS $_{\times 10^3}$ ↓ | MSE $_{\times 10^4}$ ↓ | SSIM$_{\times 10^2}$ ↑ | Whole ↑ | Edited ↑ |
| Baseline | 27.24 | 22.13 | 105.46 | 87.34 | 83.48 | 26.83 | 23.67 |
| 30 | 25.99 | 22.41 | 102.33 | 83.06 | 83.88 | 26.92 | 23.79 |
| 29 | 26.13 | 22.41 | 102.26 | 82.66 | 83.85 | 26.90 | 23.71 |
| 28 | 25.96 | 22.45 | 102.14 | 81.99 | 83.91 | 26.96 | 23.83 |
| 27 | 25.96 | 22.44 | 102.39 | 82.15 | 83.91 | 26.92 | 23.85 |
| 26 | 25.94 | 22.44 | 102.70 | 82.27 | 83.91 | 26.95 | 23.84 |

Figure 3: Qualitative comparison results with more complex prompts.

to the methods based on Flux such as RF-Inversion [19], RF-Solver [26], FireFlow [3], SplitFlow[†] outperforms the prior works. Within the SD3.5 model, SplitFlow demonstrates superior background preservation capability compared to prior works.

**Qualitative Comparison.** The qualitative comparison results are presented in Fig. 2. Across various scenarios—including *change*, *add*, and *delete* object prompts—our proposed *SplitFlow* demonstrates superior editability while effectively preserving the background. For instance, as shown in Fig. 2-(d), which involves adding "with hat" to a portrait of Mozart, other methods fail to generate the hat or distort the original image, whereas our method successfully synthesizes the hat while maintaining the integrity of the source image. Although it is well known that enhancing editability often compromises fidelity, our approach achieves a favorable balance by disentangling gradients within the flow during the editing process with decomposition. In Fig. 3, to further demonstrate the effectiveness of our method, we provide qualitative comparison results with more complex scenario. In the first row, given the source prompt "Three giraffes walk in a line through a lush, zoo-like forest path, while another animal rests near a pond," the editing prompt requires changing the three giraffes to elephants and the other animal to a tiger. While FlowEdit fails to convert "three giraffes" to "three elephants," our method successfully performs the transformation and better preserves the semantic detail of "in a line." Additional qualitative results under complex scenarios are provided in the supplementary material.

## 5.3 Detailed Analysis

**Component Analysis.** To validate the effect of the designed component, we conducted an ablation study on PIE-benchmark as shown in Table 2. The result of simply averaging the individual flows after decomposition is reported in the second column of Table 2, denoted as *AVG*. Interestingly, even naive averaging of sub-flows maintains CLIP similarity on par with the baseline [11], while significantly improving background preservation, as reflected in metrics such as PSNR, LPIPS, MSE, and SSIM. We attribute this to the semantic decoupling effect of sub-flows, which localize edits to specific attributes and reduce unintended changes in irrelevant regions. While semantic decomposition and flow separation are key contributions of our work, our overarching objective extends beyond fidelity enhancement. Our goal is to strike a balance between fidelity and editability, ensuring that complex, multi-attribute prompts are both faithfully represented and accurately reflected in the edited outputs. To this end, our proposed components—Latent Trajectory Projection (LTP) and Velocity Field Aggregation (VFA)—go beyond averaging by explicitly aligning sub-flows with the global editing direction and adaptively weighting their contributions. By applying LTP to align each

sub-flow with the target flow, we observe meaningful improvements in CLIP similarity, particularly in the *Edited* metric, which is computed over the foreground mask. Although LTP results in lower background preservation compared to simple averaging, it still outperforms the baseline in all metrics. Furthermore, when VFA is applied, each sub-flow contributes more effectively to the final trajectory. This not only enhances background preservation—similar to the averaging strategy—but also improves CLIP similarity by promoting global semantic alignment while preserving diversity across sub-prompts.

**Ablation Study on Aggregation Timestep.** In Table 3, we conduct an ablation study on the aggregation timestep $\eta_{dec}$ to evaluate the effectiveness and robustness of SplitFlow. Across all tested configurations, SplitFlow consistently outperforms the baseline. As $\eta_{dec}$ decreases—corresponding to a longer decomposition period—we observe improved editability at the expense of background preservation. Considering this trade-off, we set $\eta_{dec} = 28$ for our final configuration. Compared to the baseline, the total number of steps required by SplitFlow can be calculated as $N \times (\eta_{max} - \eta_{dec}) + \eta_{max}$. Since $N \leq 3$ in most cases, the inference steps can be approximated as $3 \times (33 - 28) + 33 = 48$. In practice, FlowEdit requires 57 minutes for inference, whereas SplitFlow takes 83 minutes to process 700 images on the PIE Benchmark. Additionally, prompt decomposition using an LLM takes approximately 20 minutes. Additional detailed ablation studies on LLM, cost analysis are provided in the supplementary material.

**Limitations** A potential limitation of our method lies in its dependence on the decomposition of the target prompt. Since the editing flows are derived from sub-prompts, the quality and characteristics of the final output can vary depending on the choice of LLMs. Although SplitFlow incurs a higher inference time than the baseline, we emphasize that the LLM serves only as a proxy to facilitate our main contribution—demonstrating that decomposing the editing process into sub-target flows can significantly improve image-editing performance—and proposing a principled method to aggregate these flows effectively. By showing that flow decomposition substantially enhances both fidelity and editability in image editing, this work also opens up new directions for future research. These include developing more effective prompt decomposition techniques using LLMs or vision-language models, as well as exploring optimization-based approaches to mitigate gradient conflicts during flow composition. Additional discussions, including analyses of extreme cases, are provided in the supplementary materials.

## 6 Conclusion

In this paper, we proposed SplitFlow, a flow decomposition and composition framework designed to address gradient entanglement and semantic conflict that arise in image editing with complex and multi-attribute target prompts. SplitFlow computes independent editing flows for each sub-target prompt and forms a unified trajectory through projection and aggregation, thereby maintaining semantic alignment while mitigating interference between attributes. To this end, we introduced two aggregation strategies: Latent Trajectory Projection (LTP), which aligns the directional components of the latent trajectory to ensure coherence with the global target semantics, and Velocity Field Aggregation (VFA), which adaptively integrates sub-target flows while preserving their semantic diversity. These components enable SplitFlow to effectively balance fidelity and editability—two often conflicting objectives in image editing. Extensive experiments on the PIE-Bench benchmark demonstrate that our method consistently outperforms existing approaches in both visual quality and prompt alignment. Our results confirm that decomposing the editing process into semantically meaningful flows and carefully reassembling them provides a promising direction for accurate, and high-quality text-guided image editing.

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
