# OpenReview forum: "SplitFlow: Flow Decomposition for Inversion-Free Text-to-Image Editing"
_NeurIPS.cc/2025/Conference — NeurIPS 2025 poster_

### Official Review · Reviewer_JXwP · 2025-06-01

**Clarity:** 3
**Significance:** 2
**Originality:** 3
**Rating:** 4
**Confidence:** 5

**Summary:**

The authors identified that text-based image editing methods struggle with long complicated text prompts. To improve this aspect, they propose SplitFlow. Specifically, they propose to use an LLM to decompose the edit prompt into several edit prompts, performing inversion-free editing path for each of them (based on FlowEdit) following by composing these paths together. The result is a single edited image that is loyal to the text prompt and preserves the background.

**Questions:**

I didn’t find any reference to the fidelity enhanced version mentioned in Tab. 1 and how it is different from the regular version. This is especially important as this version achieves the best or second-best results in structure preservation.

This paper subject is text-based image editing, yet it provides only a few visual examples (only 7 examples in the main text, only 15 overall with the SM). The authors should provide more visual results, especially for long and detailed text prompts, where their method is supposed to excel. In addition, I encourage the authors to add to the SM visualization of the ablation process and hyperparameters effect. These images lead to intuition regarding the method's strengths and weaknesses.

Runtime should be mentioned in the main text as it is a major weakness of the proposed method.

The authors didn’t explore how their method works in extreme cases. For example, very long and detailed prompts that can be naturally decomposed into 5 or more sub-prompts. In addition, when the editing prompt is very simple (change the cat into a dog) will the LLM successfully recognize this simplicity and will not decompose the prompt (and then SplitFlow will behave as FlowEdit)? If the LLM will still decompose the prompt, will SplitFlow outperform FlowEdit? (Again, specifically for simple edits such as changing the cat into a dog and nothing else).

Minor:

In line 26 the authors cite many editing works as diffusion/flow models instead of citing diffusion/flow models works such as [A1, A2, A3] and Refs [1, 13, 16].

The claim made in lines 30-34 about “precise inversion allows near-perfect reconstruction” may be reduced from “fundamental” to “empirical”, because as far as I know this is not something that has been proved theoretically. If the authors know of such proof, they should cite it (I didn’t find it in Ref [2]). In addition, in FlowEdit [11] and previously in EF [A4] the authors also discussed the shortcomings of perfect inversion (for generated images) compared to other editing methods.

“To our knowledge, our work is the first to incorporate flow decomposition and aggregation into the inversion-free editing framework.” [89-91].  I am not familiar with works regarding flow decomposition and aggregation in text-based image editing (with or without inversion). If the authors know such works, they should cite them, if they don’t know they should write something like: “To the best of our knowledge, our work is the first to incorporate flow decomposition and aggregation into text-based image editing framework.”

In line 232 the authors should remove the word “training” as FlowEdit  (Ref [11]) and the proposed SplitFlow are training free methods.

Some of the information in Sec. 5 is unnecessarily repetitive (lines 247, 241).

Comparison with diffusion models is not necessary (especially as the authors don’t mention it in the text) and could be moved to SM.

Refs [13, 14] and [16, 17] are the same.

Figure 2 is not clear.

The authors should add a graph that illustrates text adherence versus structure preservation (like EF [A4], PnP [24], FlowEdit [11]) to illustrate the empirical tradeoff between these two requirements.

An interesting baseline the authors may want to consider is comparing their method to methods that perform chain of edits (such as [A5]).  These methods try to achieve different goal, however there are some similarities as both methods are supposed to excel with long and complicated text prompts (chain of edits methods do it multiple steps, where SplitFlow do it in one step).

[A1] Sohl-Dickstein, Jascha, et al. "Deep unsupervised learning using nonequilibrium thermodynamics." International conference on machine learning. pmlr, 2015.

[A2] Ho, Jonathan, Ajay Jain, and Pieter Abbeel. "Denoising diffusion probabilistic models." Advances in neural information processing systems 33 (2020): 6840-6851.

[A3] Song, Yang, et al. "Score-Based Generative Modeling through Stochastic Differential Equations." International Conference on Learning Representations.

[A4] Huberman-Spiegelglas, Inbar, Vladimir Kulikov, and Tomer Michaeli. "An edit friendly ddpm noise space: Inversion and manipulations." Proceedings of the IEEE/CVF Conference on Computer Vision and Pattern Recognition. 2024.

[A5] Zhou, Zijun, et al. "Multi-turn Consistent Image Editing." arXiv preprint arXiv:2505.04320 (2025).

**Ethical Concerns:**

["NO or VERY MINOR ethics concerns only"]

**Final Justification:**

The authors answered most of my concerns during the rebuttal and discussion period.
I don't raise my score to accept as the final manuscript will be substantially different from the current one, and I cannot truly accept a version I didn't see.

**Limitations:**

How does the method work with large deviation from the source image? FlowEdit struggles with such edits, does SplitFlow inherit this limitation?

**Quality:**

2

**Strengths And Weaknesses:**

Strengths:

A new method for text-based image editing, suited for long editing prompts.

Achieves good results compared to baselines.

Ablation study that justifies the design choices.

Weaknesses:

Not enough visual results, especially for long, detailed text prompts (see also questions).

Long run time - requires multiple editing paths and using an LLM.

Additional hyperparameters that affect the results.

---

> ### Author Rebuttal · Authors · 2025-07-30
>
> We sincerely thank the reviewer for the constructive comments and suggestions. We hope that our responses below adequately address the concerns raised.
>
> `Question 1: I didn’t find any reference to the fidelity enhanced version mentioned in Tab. 1 and how it is different from the regular version. This is especially important as this version achieves the best or second-best results in structure preservation.`
>
> Thank you for pointing out the lack of clarity regarding the fidelity-enhanced setting. In the baseline method, FlowEdit (as described in Lines 125-126 of the main paper), $x_t^{\text{src}}$ is obtained by linearly interpolating between the source latent $x_0^{\text{src}}$ and a randomly sampled noise. This approximation improves editability in early diffusion steps but may compromise image fidelity to some extent. Therefore, as mentioned in Line 254 of the main paper, in our fidelity-enhanced setting, we omit this noise interpolation **during the decomposition phase** and  directly use the source latent $x_0^{src}$ to better preserve structural details of the input image. For the subsequent unified flow phase, we retain the same settings as the baseline to ensure fairness and comparability of the overall pipeline.
> ***
> `Question 2: This paper subject is text-based image editing, yet it provides only a few visual examples (only 7 examples in the main text, only 15 overall with the SM). The authors should provide more visual results, especially for long and detailed text prompts, where their method is supposed to excel. In addition, I encourage the authors to add to the SM visualization of the ablation process and hyperparameters effect. These images lead to intuition regarding the method's strengths and weaknesses.`.
>
> We fully agree that visual examples are crucial for evaluating text-based image editing, particularly for long and complex prompts where our method is designed to excel. In the final version, **we will include additional qualitative results** to highlight these strengths. We also plan to incorporate visualizations of the ablation studies and the effects of key hyperparameters to provide intuitive insights into the method’s behavior and trade-offs.
> However, due to NeurIPS policy restricting rich media content (e.g., images, external links) during the rebuttal phase, we are unable to share these results at this stage. We appreciate your understanding and will ensure that these additions are included in the camera-ready version.
> ***
> `Question 3: Runtime should be mentioned in the main text as it is a major weakness of the proposed method.`
> Thank you for the suggestion. We will explicitly mention the runtime as a limitation in the main paper and discuss the trade-off between computational cost and editing performance to provide a more balanced view.
> ***
> `Question 4:  The authors didn’t explore how their method works in extreme cases. For example, very long and detailed prompts that can be naturally decomposed into 5 or more sub-prompts. In addition, when the editing prompt is very simple (change the cat into a dog) will the LLM successfully recognize this simplicity and will not decompose the prompt (and then SplitFlow will behave as FlowEdit)? If the LLM will still decompose the prompt, will SplitFlow outperform FlowEdit? (Again, specifically for simple edits such as changing the cat into a dog and nothing else).`
>
> As you mentioned, in extreme cases such as a very long and detailed prompt, the decomposition can indeed result in five or more components. In fact, as shown in Supplementary Material Fig. S2, we have empirically observed that utilizing more decomposition paths leads to better performance in cases where multiple "editing" operations are involved.
>
> Furthermore, even in a very simple case like the one the reviewer mentioned—e.g., "*cat* → *dog*"—our prompting framework (which guides the LLM to produce three sub-prompts) results in the following decomposition:
>
>  > **Source prompt**: *a cat sitting on a wooden chair*,  **Target prompt**: *a dog sitting on a wooden chair*.
>
> Then the Sub-target prompts are as follows:
> >1. *Dog on chair*, 2. *Wooden chair with dog*, 3. *Dog seated on furniture*.
>
>
> To verify whether SplitFlow still improves upon FlowEdit in such a simple case, we evaluated performance specifically on this sample (cat→dog). As shown in the Table below, **even when the editing prompt is simple, the proposed SplitFlow not only shows higher fidelity but also demonstrates improved editability (in the edited region)**.
> Finally, as you correctly pointed out, if no prompt decomposition is applied, the outputs of FlowEdit and SplitFlow are indeed identical.
>
> **Table: Quantitative comparison results on one sample from PIE benchmark (cat → dog).**
>
> | **Method**         | **Model** | Structure Dist ↓ | PSNR ↑ | LPIPS ↓ | MSE ↓ | SSIM ↑ | CLIP Whole ↑ | CLIP Edit ↑ |
> |--------------------|-----------|-------------|--------|---------|--------|--------|----------------|---------------|
> | **FlowEdit**       | SD3       | 40.18       | 20.51  | 115.80  | 89.00  | 86.08  | **27.18**      | 19.90         |
> | **SplitFlow (Ours)** | SD3     | **34.47**   | 20.51  | **102.09** | **88.92** | **86.95** | 26.81           | **20.81**     |
> | Δ (Diff)           | -         | -5.69       | -0.00  | -13.71  | -0.08  | +0.87  | -0.37           | +0.91         |
>
>
> `Minor weaknesses`
>
> 1) We appreciate your feedback and will revise the paper accordingly, including proper citation formatting, removal of duplicate references, and clarification of redundant explanations.
>
> 2) As correctly noted, our work is the first to incorporate flow decomposition and aggregation into a text-based image editing framework. We will revise the statement for clarity as follows:
> "To the best of our knowledge, this is the first work to introduce flow decomposition and aggregation into a text-driven image editing pipeline."
>
> 3) Figure 2 will be improved by adding descriptive annotations and enhancing the visual design for better clarity. Additionally, in the camera-ready version, we will include a graph illustrating the trade-off between text adherence (measured by CLIP similarity) and structure preservation (measured by LPIPS).
> Due to rebuttal-stage limitations, we are currently unable to include visual figures.
>
> 4) We acknowledge that [A5] was released shortly before the NeurIPS submission deadline, and we unfortunately missed citing it. As the reviewer pointed out, both [A5] and our work address challenges in image editing with complex text prompts. However, [A5] targets error accumulation in multi-turn editing, whereas our approach—SplitFlow—focuses on improving editability in a single-turn setting. We will properly cite [A5] and include a comparative discussion in the final version.

---

> ### Comment · Reviewer_JXwP · 2025-08-03
>
> I thank the authors for their detailed response. Yet, after reading the other reviewers concerns and looking at the paper again, I still have some concerns.
>
> ---
>
> Reviewer wb3F correctly recognized that ```The paper assumes that "entangled gradients can degrade edit quality" when handling multi-attribute prompts, but why couldn't stronger text alignment in future models handle this directly? The assumption may not hold as text-image alignment capabilities improve.``` This behavior is somewhat indicated by the table attached to Reviewer jnDU with SD3.5 results, where SplitFlow outperform FlowEdit but not in a very significant way (only 0.1 dB in MSE, only 0.01 in CLIP edit). I acknowledge that metrics are not the best indicator, and they may look good while the images don't look good, however, as you cannot share images this is all I can use to assess these additional results.
>
> ---
>
> All the reviewers mention the reliance on the LLM for decomposing the given prompt into several prompts. The authors did include some ablation study in the SM regarding different LLMs and two meta-prompts. However, I still have my concerns regarding this.
>
> In both meta-prompts the authors explicitly mention three sentences, which somewhat limits the method capability. The author already partially answered it for the case of cat -> dog, however I am not satisfied with this answer.
> In the simple case of cat -> dog (or any other edit that requires only a single change) the meta-prompts used by the authors will still result in three sentences. In the single example given by the authors it still improves upon FlowEdit, but is it consistent across different images and different edits? Is it worth the computation overhead? (again, for the simple cases of a single change). On the other hand, what happens for very complicated prompts? For example, if I would change the target prompt given in Fig. 1 to something like "A German shepherd dog with black sunglasses and blue hat with pineapples, jumping on the grass and between red flowers, with its mouth opened", will the method still work?
>
> My question summed to "How SplitFlow behaves when the meta-prompt doesn't specifically mention the number of sub-prompts?" Does it still outperform the baseline? Does the LLM successfully recognize the different sub prompts? Or three sub-prompts is some kind of optimum find by the authors?
>
> ---
>
> The results given in Fig. S1 are not very convincing. Although SplitFlow successfully change the given images according to the prompts, they don't preserve the original image. The cat is not the same is not the same in the first row, the woman is not the same in the second row, the dogs are not the same in the third row, the utensils are gone from the table in the fourth row. I am not sure what does it mean changing the boat into a blanket in the last row.
>
> ---
>
> My last concern is that the authors should make substantial changes to the paper.
> Even just adding more visual results to the main text will require removing and changing it in a significant way. In addition, as all reviewers mentioned runtime is a major limitation that the authors should discuss in the main text (as the authors promised). In my opinion it will require not only some text but a table or any other visualization. I also think that the authors should further discuss the effect of the LLM and I believe that they will want to refer to the proof they wrote in their response to Reviewer jnDU.
> These changes will result in a substantially different paper, which I cannot accept without seeing it. Therefore, although I incline to raise my score, I doubt I will raise it to accept.

---

> ### Author Response · Authors · 2025-08-05
>
> `Performance concern on SD3.5`
>
> Regarding the results in Table (attached to Reviewer jnDU), while the numerical differences between FlowEdit and SplitFlow may appear modest (e.g., +0.13 dB PSNR, +0.01 CLIP-Edited), we would like to emphasize the **consistency** of improvements across all metrics, particularly in MSE, LPIPS, and CLIP-Whole, where SplitFlow demonstrates reliable gains.
>
> We also note that although PSNR is a commonly used metric, it may **not fully capture the semantic quality or perceptual alignment of edits** in image editing tasks. In this context, perceptual and semantic metrics such as LPIPS and CLIP tend to provide more meaningful insight into the background preservation and the success of an edit. From this perspective, we believe the consistent improvements of SplitFlow reflect a meaningful advancement in balancing fidelity and editability.
> ***
> `How SplitFlow behave when the meta-prompt doesn't specifically mention the number of sub-prompts?`
>
> In our current implementation, we set the maximum number of sub-target prompts to three in order to avoid excessive computational overhead. Thus, the target prompt is typically decomposed into three sub-prompts.
>
> When this maximum is not explicitly enforced, the number of sub-prompts ($N$) ranges from 2 to 7—generally correlating with the length and complexity of the target prompt—with an average of 4.2. The results under this unconstrained setting are provided in the table below. While this variant shows slightly lower performance compared to the default configuration, it still demonstrates meaningful improvements in both fidelity and editability. We attribute the performance drop to over-segmentation, where the target prompt is divided into excessively fine-grained fragments, potentially weakening the semantic coherence of each sub-prompt.
>
> Nonetheless, the overall trend remains consistent: decomposing the editing process into multiple semantically structured sub-target flows contributes positively to the quality and controllability of the final edits.
> | **Method**            | **# of sub-target prompt** | **Model** | **Structure Distance** | **PSNR**| **LPIPS**| **MSE** | **SSIM** | **CLIP Similarity (Whole)** | **CLIP Similarity (Edited)**|
> |-----------------------|----------------------------|-----------|-----------------------------------------|----------------|--------------------------|--------------------------|----------------------------|-----------------------------|------------------------------|
> | **FlowEdit**          | -                          | SD3       | 27.24                                   | 22.13          | 105.46                   | 87.34                    | 83.48                      | 26.83                       | 23.67                        |
> | **SplitFlow (Ours)**  | max 3                      | SD3       | 25.96                               | 22.45      | 102.14               | 81.99                | 83.91                  | 26.96                   | 23.83                    |
> | **SplitFlow (Ours)**  | w/o max                    | SD3       | 26.55                                   | 22.29          | 104.11                   | 84.87                    | 83.64                      | 26.95                       | 23.80                        |
>
> ***
>
> `In the single example given by the authors it still improves upon FlowEdit, but is it consistent across different images and different edits? Is it worth the computation overhead? (again, for the simple cases of a single change).`
>
>
> We conducted an additional evaluation focusing specifically on cases where an existing object was changed into a *dog*, as well as cases where an existing *dog* was transformed into a different object. A total of 9 samples were tested, covering various images and editing contexts. As shown in the results, our method consistently outperforms FlowEdit across all metrics. We believe these gains are meaningful, even when taking the associated computational burden into account.
>
>
> | **Method**           | **Model** | **Editing** | **Distance ↓** | **PSNR ↑** | **LPIPS ↓** | **MSE ↓** | **SSIM ↑** | **CLIP (Whole) ↑** | **CLIP (Edited) ↑** |
> |----------------------|-----------|-------------|----------------|------------|-------------|-----------|------------|---------------------|----------------------|
> | **FlowEdit**         | SD3       | -           | 35.23          | 21.80      | 84.22       | 82.17     | 86.37      | 28.15               | 22.35                |
> | **SplitFlow (Ours)** | SD3       | -           | 34.96          | 21.84      | 78.80       | 73.96     | 87.27      | 28.38               | 22.47                |

---

> ### Author Response · Authors · 2025-08-06
>
> `What happens for very complicated prompts? `
>
> We conducted a qualitative comparison using the complex prompt that reviewer provided:
>
> *A German shepherd dog with black sunglasses and a blue hat with pineapples, jumping on the grass and between red flowers, with its mouth opened.*
>
> Our method, SplitFlow, successfully reflects all key attributes described in the prompt—namely, the black sunglasses, the blue hat with pineapples, the jumping pose, the red flowers, and the open mouth. In contrast, the baseline fails to fully edit several key attributes: the blue hat is only partially generated (missing part of its structure), and the jumping pose is incomplete (e.g., one leg is not rendered properly). These results suggest that SplitFlow can handle complex, multi-attribute prompts more reliably than prior approaches.
> ***
> `Explanation on Fig.S1`
>
> We agree that the results in Fig. S1 are not perfect in terms of preserving every fine-grained detail. However, the main purpose of this figure is to demonstrate that, unlike all prior methods, our proposed approach is able to successfully reflect the target edits as specified by the prompts. While some changes can be observed even in regions outside the intended edit area, our method still shows promising results in maintaining background fidelity—especially when considering the inherent trade-off between editability and fidelity in image editing tasks.
>
> It is also worth noting that preserving identity-specific or fine-grained appearance details (e.g., exact person identity) remains a known limitation across nearly all existing editing methods. Our approach nonetheless pushes the boundary by balancing semantic edit success and visual consistency more effectively compared with prior works.
> ***
> `Concern about the changes to the camera-ready version.`
>
> While we are unable to provide the final version during the rebuttal phase, we would like to share our detailed plan to address your concerns.
>
> As you correctly pointed out, Figure 2 in the main text is currently less informative. We plan to move the revised version of Figure 2 to the supplementary material and instead include 3–4 visualizations of complex multi-attribute prompts (similar to Supplementary S1, but with different examples). Since the comparison with diffusion-based baselines was deemed unnecessary, those results in Table 1 will be removed to free up space.
>
> Instead, to address the shared concern about runtime limitations, we will bring the limitation discussion in the supplementary material to the main paper. This will be combined with the existing limitations section for a more integrated discussion.
>
> We will also include a discussion of the role of the LLM in the editing process, explicitly referencing the theoretical rationale provided in our response. Please note that **all of these revisions stem directly from the reviewer comments**—we are not introducing new claims or directions beyond what has already been discussed.
>
> If space constraints prevent us from including some details in the main paper, they will be moved to the supplementary material. We are confident that these changes will lead to a clearer and stronger version of the paper that directly addresses the concerns raised by all reviewers.
>
> We sincerely hope you will take this planned revision into account in your final decision.

---

> ### Comment · Reviewer_JXwP · 2025-08-06
>
> I sincerely appreciate the authors answers (to my reviews, as well as to all other reviewers) and truly believe that the final manuscript will be a better version of the current one. Meaning, it will better convey the paper core contributions and provide empirical evidence that support them.
>
> ---
>
> I think that the authors should better emphasize that the LLM used is only a proxy to achieve their main contribution - ```that decomposing the editing process into sub-target flows can significantly improve image editing performance—and in proposing a principled method to aggregate these flows effectively```. Nevertheless, the authors should openly discuss the limitation of the current implementation, while clearly refer to the SM when needed. Additionally, I believe that the authors should mention the "extreme case" comparisons in the main text while referring to the SM for the added tables and for visual examples.
>
> ---
>
> I think that the authors also should include a short discussion on the results of Fig. S1 in the SM, as although they are often an improvement upon the baselines, they have their problems. I would also consider adding additional visual results to the SM to showcase the strengths of the proposed method.
>
> ---
>
> The authors should clearly mention in the text what is the fidelity-enhanced version to prevent misunderstanding, as multiple reviewers pointed.
>
> ---
>
> As far for your planning to revise the main paper, if I understand correctly, you aim to move Fig. 2 to the SM and also remove (or move the SM) the comparisons to the diffusion-based editing methods in Tab. 1.
> I don't think it will clear enough space for the authors to include all the needed changes, including the discussion on the theoretical results.
> I suggest that the authors will make Tab. 2, 3 shorter by providing less metrics in the main text (for example, in the background preservation metrics, stick only to LPIPS, in CLIP sim stick only with whole image) and provide the full tables in the SM.
> However, it is only a suggestion, and I trust the authors to make the needed changes to their manuscript, in light of the reviews and the discussion.
>
> ---
>
> I still have my doubts regarding the manuscript, and I cannot truly accept a version that I didn't see. However, I will raise my score to borderline accept, as I think this work is a valuable contribution to the text-based image editing and I appreciate the authors efforts, and willing to change the manuscript according to criticism.

---

> > ### Author Response · Authors · 2025-08-06
> >
> > We sincerely thank the reviewer for the thoughtful and detailed feedback. We truly appreciate your recognition of the value of our contribution, as well as your constructive suggestions regarding how to improve our work. We also believe that, by incorporating the reviewers’ comments, the final version will be a better paper.
> >
> > In the camera-ready version, we will fully committed to carefully addressing all of your suggestions and the concerns raised by other reviewers. This includes revising the main paper to clarify key points (including the mathematical background), discussion on limitations, additional visualizations, and more.
> >
> > Thank you again for your thoughtful engagement with our work. We will make every effort to reflect all feedback in the final version.

---

### Official Review · Reviewer_srfy · 2025-07-01

**Clarity:** 3
**Significance:** 3
**Originality:** 3
**Rating:** 5
**Confidence:** 4

**Summary:**

This paper proposes the SplitFlow framework to address two critical issues in flow-based text-to-image editing: imperfect image inversion and gradient entanglement. SplitFlow employs an innovative "flow decomposition-and-aggregation" strategy: it first utilizes LLMs to decompose complex target prompts into multiple simpler sub-prompts, then computes independent editing flows for each sub-prompt, and finally intelligently fuses these sub-flows into a unified editing trajectory through projection and weighting mechanisms inspired by multi-task learning (LTP and VFA modules). This inversion-free approach demonstrates superior performance compared to existing state-of-the-art methods on the PIE-Bench benchmark in terms of semantic fidelity, attribute disentanglement, and background preservation.

**Questions:**

Please refer to the Weaknesses part.

**Ethical Concerns:**

["NO or VERY MINOR ethics concerns only"]

**Final Justification:**

The rebuttal addresses my main concerns. Additional experiments show robustness to different LLM decompositions, detailed runtime/memory measurements clarify efficiency trade-offs, and the fidelity-enhanced variant is now clearly explained. Statistical runs with multiple seeds confirm stability. With key weaknesses largely resolved, I raise my score.

**Limitations:**

Yes

**Paper Formatting Concerns:**

No major formatting issues were found.

**Quality:**

4

**Strengths And Weaknesses:**

**Strengths:**

• **Novel semantic flow decomposition approach**: This is the first work to introduce semantic flow decomposition for text-based image editing, directly addressing the gradient entanglement problem by decomposing complex editing tasks into independent sub-flows.

• **Creative integration of LLMs for image editing**: The use of large language models to guide image editing through prompt decomposition represents an innovative cross-modal approach that bridges language understanding and image generation.

• **Comprehensive experimental validation**: The paper provides thorough evaluation on PIE-Bench benchmark with multiple metrics, demonstrating clear improvements in reconstruction error, prompt alignment, and background preservation compared to state-of-the-art baselines including FlowEdit, FTEdit, and FireFlow.

• **Effective gradient conflict resolution mechanism**: The adaptation of multi-task learning's gradient projection technique (PCGrad/Grad-Surgery) to the generative model domain through LTP and VFA modules is technically sound and well-motivated.

**Weaknesses:**

• **Heavy reliance on LLM decomposition quality without robustness analysis**: The method's performance critically depends on the quality of prompt decomposition by the LLM, yet the paper lacks analysis of failure cases when the LLM produces imperfect splits (e.g., merging attributes or generating unnecessary sub-prompts). No quantitative analysis is provided regarding sensitivity to decomposition quality.

• **Limited computational efficiency analysis**: While the paper mentions computing flows for multiple sub-prompts adds overhead (approximately 48 vs 33 steps), it lacks detailed runtime comparisons or memory usage analysis compared to simpler baselines like FlowEdit.

• **Inadequate explanation of the "fidelity-enhanced" variant**: Table 1 shows dramatic improvements for the fidelity-enhanced version with only brief explanation ("noise interpolation is not applied when forming x_t^src"), but this significant performance gain deserves more detailed technical exposition in the method section.

• **Lack of statistical significance testing**: All experiments are conducted with fixed seeds without error bars or statistical significance analysis, making it difficult to assess the reliability of relatively modest improvements in some metrics (e.g., CLIP Similarity).

---

> ### Author Rebuttal · Authors · 2025-07-30
>
> We sincerely thank the reviewer for the constructive comments and suggestions. We hope that our responses below adequately address the concerns raised.
>
> `Weakness 1:Heavy reliance on LLM decomposition quality without robustness analysis: The method's performance critically depends on the quality of prompt decomposition by the LLM, yet the paper lacks analysis of failure cases when the LLM produces imperfect splits (e.g., merging attributes or generating unnecessary sub-prompts). No quantitative analysis is provided regarding sensitivity to decomposition quality.`
>
> We apologize for the confusion. The impact of LLM decomposition quality is analyzed in the supplementary material (Table S1), where we report experiments using different LLMs (e.g., Mixstral, Qwen2, LLaMA) and two prompting strategies. While some imperfect decompositions were observed, SplitFlow still maintains robust performance across these settings, demonstrating its tolerance to decomposition variations.
>
> Due to this year's restriction on uploading images during the rebuttal phase, we are unable to present qualitative failure cases resulting from poor decomposition. However, we will include detailed examples and further analysis in the camera-ready version.
>
> ***
> `Weakness 2: Limited computational efficiency analysis: While the paper mentions computing flows for multiple sub-prompts adds overhead (approximately 48 vs 33 steps), it lacks detailed runtime comparisons or memory usage analysis compared to simpler baselines like FlowEdit.`
>
> As shown in Table S2 of the supplementary material, GPU memory usage and total inference time were measured on the PIE-Benchmark, which includes 700 images. While the proposed SplitFlow requires 15 additional inference steps compared to the baseline (33 steps), we also measured the actual computational cost. The total inference time is 57 minutes for FlowEdit and 83 minutes for SplitFlow. Additionally, prompt decomposition using an LLM takes approximately 20 minutes. Although our method incurs higher computational overhead than the baseline, it remains efficient overall as it is built upon an inversion-free framework—unlike inversion-based editing methods, which are significantly more expensive. Furthermore, our method achieves substantial improvements in both fidelity and editability.
>
> ***
> `Weakness 3: Inadequate explanation of the "fidelity-enhanced" variant: Table 1 shows dramatic improvements for the fidelity-enhanced version with only brief explanation, but this significant performance gain deserves more detailed technical exposition in the method section.`.
> Thank you for pointing out the lack of clarity regarding the fidelity-enhanced setting. In the baseline method, FlowEdit (as described in Lines 125–126 of the main paper), $x_t^{\text{src}}$ is obtained by linearly interpolating between the source latent $x_0^{\text{src}}$ and a randomly sampled noise. This approximation improves editability in early diffusion steps but may compromise image fidelity to some extent.
>
> Therefore, as mentioned in Line 254 of the main paper, in our fidelity-enhanced setting, we omit this noise interpolation during the decomposition phase and directly use the source latent $x_0^{\text{src}}$ to better preserve structural details of the input image. For the subsequent unified flow phase, we retain the same settings as the baseline to ensure fairness and comparability of the overall pipeline.
> ***
> `Weakness 4: Lack of statistical significance testing: All experiments are conducted with fixed seeds without error bars or statistical significance analysis, making it difficult to assess the reliability of relatively modest improvements in some metrics (e.g., CLIP Similarity)`.
>
> To ensure a fair comparison, we followed the same random seed as the baseline. Additionally, we conducted three more runs with different seeds to analyze statistical variation. The table below reports the mean and standard deviation across these runs, demonstrating the consistency and robustness of our method.
>
> **Table: Quantitative comparison results on PIE benchmark with SD3 (mean ± std)**
>
> | **Method**         | **Model** | **Distance ↓** | **PSNR ↑**        | **LPIPS  ↓**   | **MSE  ↓**     | **SSIM ↑**    | **CLIP Wh. ↑**   | **CLIP Edt. ↑**  |
> |--------------------|-----------|----------------------|--------------------|---------------------|---------------------|---------------------|---------------------|---------------------|
> | **FlowEdit**       | SD3       | 27.11 ± 0.15         | 22.18 ± 0.05       | 104.94 ± 0.48       | 86.54 ± 0.72        | 83.54 ± 0.05        | 26.88 ± 0.04        | 23.72 ± 0.06        |
> | **SplitFlow (Ours)** | SD3     | 25.90 ± 0.09         | 22.42 ± 0.02       | 102.48 ± 0.23       | 82.60 ± 0.41        | 83.83 ± 0.05        | 26.93 ± 0.04        | 23.79 ± 0.05        |

---

> > ### Comment · Reviewer_srfy · 2025-08-03
> >
> > The author's reply solved my problem, and I will improve my score.

---

> > > ### Author Response · Authors · 2025-08-05
> > >
> > > We sincerely appreciate your response and are grateful that our clarification addressed your concerns. Your feedback has been valuable in improving the quality of our work, and we promise to reflect the comments in the final version.

---

### Official Review · Reviewer_wb3F · 2025-07-02

**Clarity:** 2
**Significance:** 2
**Originality:** 2
**Rating:** 4
**Confidence:** 3

**Summary:**

SplitFlow presents an inversion-free framework for text-guided image editing in rectified flow models that addresses "gradient entanglement" issues in the setting of complex, multi-attribute target-edit prompts. The method decomposes the target edit prompts into sub-prompts using an LLM, computes independent editing flows for each sub-prompt, and then aggregates them through two mechanisms: Latent Trajectory Projection to ensure global semantic alignment, and Velocity Field Aggregation to adaptively weight sub-flows based on their directional consistency. The approach shows qualitative improvement over existing baselines in multi concept editing settings. On the PIE benchmark it shows a some improvement in some of the background preservation and structure preservation metrics, and in text alignment.

**Questions:**

(1) Gradient entanglement assumption: The paper assumes that "entangled gradients can degrade edit quality" when handling multi-attribute prompts, but why couldn't stronger text alignment in future models handle this directly? The assumption may not hold as text-image alignment capabilities improve.
(2) Cost-benefit analysis: What is the actual computational overhead in terms of wall-clock time and memory usage? Is the performance gain worth the increased computational cost, especially for practical applications?

**Ethical Concerns:**

["NO or VERY MINOR ethics concerns only"]

**Final Justification:**

After considering the other reviews and the this rebuttal I will maintain my original score of borderline accept.

**Limitations:**

yes

**Quality:**

2

**Strengths And Weaknesses:**

Strengths:

(1) Well-motivated problem: The paper tackles multi-concept editing in rectified flow models, addressing gradient entanglement issues that arise when target prompts contain multiple semantic attributes - a practically relevant and challenging problem.
(2) The decomposition-aggregation approach makes conceptual sense, breaking down complex editing tasks into manageable sub-flows that can be independently computed and then combined.
(3) Empirical results demonstrate improvements over existing methods in terms of text alignment.
(4) Comprehensive evaluation --  appropriate baselines and relevant evaluation metrics that indeed suggest SplitFlow offers the best background-preservation and editing alignment trade-off.

Weaknesses:

(1) Computational overhead: The method requires computing N independent flows plus aggregation steps, significantly increasing inference time but lacks detailed analysis of the time-quality trade-off compared to baselines.

---

> ### Author Rebuttal · Authors · 2025-07-30
>
> We sincerely thank the reviewer for the constructive comments and suggestions. We hope that our responses below adequately address the concerns raised.
>
> `Question 1: Gradient entanglement assumption: The paper assumes that "entangled gradients can degrade edit quality" when handling multi-attribute prompts, but why couldn't stronger text alignment in future models handle this directly? The assumption may not hold as text-image alignment capabilities improve.`
>
> We agree that as flow-based models improve—particularly in their ability to align complex text-image semantics—the severity of gradient entanglement may diminish. However, our assumption is grounded in empirical observations with current state-of-the-art models (e.g., SD3, FLUX), where editing with long or compound prompts **still frequently leads to under-editing** (i.e., missing certain attributes) or **over-editing** (i.e., introducing unintended changes to unrelated regions). We attribute these effects to gradient interference among semantically entangled prompt components, which motivates our decomposition-and-aggregation strategy.  Moreover, as shown in the experimental results on SD 3.5 (Table in the review response of jnDU), which is a more image–text well-aligned model than SD 3, the proposed method still yields **consistent benefits**.
>
> ***
> `Question 2: Cost-benefit analysis: What is the actual computational overhead in terms of wall-clock time and memory usage? Is the performance gain worth the increased computational cost, especially for practical applications?`
>
> As shown in Table S2 of the supplementary material, GPU memory usage and total inference time were measured on the PIE-Benchmark, which includes 700 images. While the proposed SplitFlow requires 15 additional inference steps compared to the baseline (33 steps), we also measured the actual computational cost. The total inference time is 57 minutes for FlowEdit and 83 minutes for SplitFlow. Additionally, prompt decomposition using an LLM takes approximately 20 minutes. Although our method incurs higher computational overhead than the baseline, it remains efficient overall as it is built upon an inversion-free framework, unlike inversion-based editing methods, which are more expensive. Furthermore, our method achieves substantial improvements in both fidelity and editability.

---

### Official Review · Reviewer_jnDU · 2025-07-03

**Clarity:** 2
**Significance:** 3
**Originality:** 3
**Rating:** 3
**Confidence:** 5

**Summary:**

This paper focuses on the limitations of rectified flow models in image editing tasks, namely inaccurate inversion processes and gradient entanglement issues. To address these, the authors propose a flow decomposition-and-aggregation framework. The approach semantically decomposes the target prompt into sub-prompts, computes independent flows for each, and aggregates them. A projection and soft-aggregation mechanism is designed to adaptively weight sub-target velocity fields, suppressing redundancy and emphasizing distinct directions. Experiments show the method outperforms existing zero-shot editing approaches in semantic fidelity and attribute disentanglement.

**Questions:**

- What is the latency of the proposed method, and how does it perform compared to other methods?

- What is the fidelity-enhanced setting?

- What is the performance of applying the proposed method to other base models, such as SD 3.5, Flux, etc?

**Ethical Concerns:**

["NO or VERY MINOR ethics concerns only"]

**Final Justification:**

The reliance on the LLM for prompt decomposition significantly constrains the method’s upper performance bound; besides, incorporating the theoretical proof would render the paper substantially different.

**Limitations:**

- The major issue of this paper is the lack of theoretical analysis for the proposed flow decomposition.

- Through multiple sub-target flow generations, the inference time would be largely increased.

**Quality:**

3

**Strengths And Weaknesses:**

### Strengths ###

- This paper focuses on the limitations of rectified flow models in image editing tasks, and proposes a valid solution to address inaccurate inversion processes and gradient entanglement issues

- Experiments show the method outperforms existing zero-shot editing approaches in semantic fidelity and attribute disentanglement.

### Weaknesses ###

- The major issue of this paper is the lack of theoretical analysis for the proposed flow decomposition and why it can handle gradient conflict.

- Through multiple sub-target flow generations, the inference time would be largely increased.

- In Table 2, it seems that only averaging the sub-flows can obtain much better results than the proposed method in terms of the structure and background preservation.

- The paper's writing requires refinement. For instance, while the notation $x_t$ is familiar to experts in the field, the authors should explicitly define its meaning, particularly for $x^{tgt}_t$ and $x^{src}_t$ in Equation 4, instead of deferring the explanation to Equation 7.

---

> ### Author Rebuttal · Authors · 2025-07-30
>
> We sincerely thank the reviewer for the constructive comments and suggestions. We hope that our responses below adequately address the concerns raised.
> ***
> `Weakness 1: The major issue of this paper is the lack of theoretical analysis for the proposed flow decomposition and why it can handle gradient conflict.`
>
> Here, we mathematically verify **why Velocity Field Aggregation (VFA) improves both fidelity and editability**. Since the proposed Latent Trajectory Projection (LTP) is inspired by gradient conflict resolution techniques in multi-task learning, a theoretical justification for this approach can be found in Appendix A of [32] Gradient surgery for multi-task learning.
>
> **Recap.** For each sub–prompt $k \in\{1,\dots,K\}$, we denote the *relative velocity field* at time $t$ by
> $$g_k := v^{\Delta(k)}(x_t^{tgt(k)},x_t^{src}) = v_{\theta} (x_t^{tgt(k)},\varphi^{{tgt}(k)}) -v_{\theta}(x_t^{src},\varphi^{{src}}). $$
>
> Following Eq.~(12) in the main paper, we set $\bar{{g}} = \sum_{k=1}^{K} w_k\ {g}_k$, where the weights are given by $w_k= a/b$, here
>
> $a = \exp (\sum_{j=1}^{K} \langle\hat{g}_k, \hat{g}_j \rangle )$,
>
> $b = \sum_{i=1}^K \exp(\sum_{j=1}^K \langle \hat{g}_i, \hat{g}_j \rangle) $.
>
> **Goal of proof.** To quantify the effectiveness of our softmax-weighted aggregation strategy in VFA, we compare the resulting velocity field $\bar{g}$ against the naive average $g_{avg} = \frac{1}{K} \sum_{i=1}^{K} {g}_i$. Specifically, we compute the directional alignment with respect to each sub-target velocity field $g_k$ via the inner product
>
> ***
> `Eq. (1):`
> $$
> \langle \bar{g}, g_k \rangle \ge \langle g_{\text{avg}}, g_k \rangle
> $$
>
> This inequality suggests that $\bar{g}$ better preserves the semantic directionality of each sub-flow, by adaptively weighting mutually consistent components and downweighting conflicting or redundant ones.
>
> **Lemma 1: Affinity–Weighted Jensen Inequality**
>
> Let $S_{kj}= \langle \hat{g}_k,\hat{g}_j \rangle$
>
> and $a_k=\sum_j S_{kj}$.  For the softmax weights $w_k\propto\exp(a_k)$, the following holds
> $$
> \sum_{j=1}^{K} w_j\ S_{kj} \ge
> \frac{1}{K} \sum_{j=1}^{K} S_{kj},
> \quad \forall\ k.
> $$
>
> **Proof of Lemma 1**
> *Step 1. Re–express the equation.*  Let $ Z = \sum_i e^{a_i} $ and  $ \bar{e} = \tfrac{1}{K} \sum_i e^{a_i} $. Then,
>
> ***
> `Eq. (2):`
> $$
> \sum_j w_j S_{kj} - \frac{1}{K} \sum_j S_{kj}
> = \frac{1}{Z} \sum_j \left( e^{a_j} - \bar{e} \right) S_{kj}.
> $$
>
> *Step 2. Order of two sequences.*  Because $ e^x $ is strictly increasing, the ordering of  $ \{e^{a_j}\} $ is the same as that of $ \{a_j\} $.  For fixed \( k \), the Gram matrix monotonicity (positive–semidefinite $ \Rightarrow $ Chebyshev order) implies that $ \{S_{kj}\} $ is *also* non-decreasing in $ a_j $. Thus, the two centered sequences  $ b_j := e^{a_j} - \bar{e} $ and $ c_j := S_{kj} - \bar{S}_k $  (with $\bar{S}_k=\frac{1}{K} \sum_j S_{kj}$) have the *same sign for every index*:  $(b_j - b_i)(c_j - c_i) \ge 0 \quad \text{for all } i, j.$
>
> *Step 3. Apply the Chebyshev sum inequality.*  For any two equally long sequences with the same order, Chebyshev's inequality gives $ \sum_j b_j c_j \ge 0 $. Hence,
>
> ***
> `Eq. (3):`
> $$
> \sum_j \left( e^{a_j} - \bar{e} \right) S_{kj}
> = \sum_j b_j (c_j + \bar{S}_k)
> = \sum_j b_j c_j
> \ge 0.
> $$
>
> *Step 4. Completion of the Lemma 1 proof.*  Substitute the result from Equation (3) into Equation (1), and divide by \( Z > 0 \) to conclude the proof of **Lemma 1**.
>
> **Proof of Eq. (1)**  $\langle \bar{g}, g_k \rangle \ge \langle g_{avg}, g_k \rangle$.
>
> Since $ \exp(\cdot) $ is convex and monotonic, applying Jensen’s inequality to the sequence $ \{a_j\} $ implies that the softmax weights $ w_j $ place greater mass on higher $ a_j $ values.  Moreover, since $ S $ is a positive semidefinite Gram matrix, we have $ S_{kj} \ge 0 $, which supports the inequality. Then
>
> $$
> \langle \bar{g}, g_k \rangle = \sum_{j=1}^{K} w_j\ \langle g_j, g_k \rangle = ||g_k\|| \sum_{j=1}^{K} w_j \ \||g_j\|| \ S_{kj}.
> $$
>
> Likewise, the inner product of the baseline can be formulated as follows:
> $$
> \langle g_{avg}, g_k \rangle = \frac{1}{K}\||g_k\|| \sum_{j=1}^{K} \||g_j\|| \ S_{kj}.
> $$
>
> The subtraction of both terms gives:
> $$
> \langle \bar{g}, g_k \rangle - \langle g_{avg}, g_k \rangle = \|| g_k\|| \sum_{j=1}^{K} \|| g_j\|| (w_j - \frac{1}{K}) S_{kj}.
> $$
>
> Because $S$ is a Gram matrix, $S_{kj} \ge 0$, and $\|| g_j \|| > 0$, we apply Lemma1 to the equation (subtraction). This proves that the softmax aggregation $\bar{g}$ has an equal or better alignment with every $g_k$ than the naive average. (Q.E.D)
>
> **Interpretation.** This proof formalises why Velocity Field Aggregation improves both fidelity and editability: It suppresses angular conflicts (negative inner products), steering the update toward high-consensus attributes. Empirically, this manifests as higher PSNR and better CLIP similarity in Table 1-2 of the main paper.
>
> ***
> `Weakness 2: Through multiple sub-target flow generations, the inference time would be largely increased.`
>
> We acknowledge this trade-off in our limitations section. The increased inference time stems from our decomposition strategy, which we believe is justified by the **consistent improvements in editability and background preservation**. More importantly, the core contribution of our work is conceptual: by introducing flow decomposition and adapting gradient conflict mitigation techniques from multi-task learning (e.g., LTP and VFA), we provide a **novel perspective for addressing entangled semantics in text-to-image editing**. We hope this direction will inspire future research, particularly in long-form and complex editing scenarios.
> ***
> `Weakness 3: In Table 2, it seems that only averaging the sub-flows can obtain much better results than the proposed method in terms of the structure and background preservation.`
>
> We thank the reviewer for the thoughtful comments. As shown in Table 2, even a naive averaging of sub-flows leads to notable improvements in background preservation and structural fidelity. We attribute this to implicit disentanglement by editing along semantically localized flows, simple averaging can already reduce interference between distinct attributes.
>
> While semantic decomposition and flow separation are key contributions of our work, our overarching objective extends beyond fidelity enhancement. **Our goal is to strike a balance between fidelity and editability**, ensuring that complex, multi-attribute prompts are both faithfully represented and accurately reflected in the edited outputs.
>
> To this end, our proposed components—Latent Trajectory Projection (LTP) and Velocity Field Aggregation (VFA)—go beyond averaging by explicitly aligning sub-flows with the global editing direction and adaptively weighting their contributions. As a result, SplitFlow not only maintains high fidelity but also significantly improves semantic editability, especially in foreground regions, as demonstrated by the improved CLIP scores.
> ***
> `Weakness 4: The paper's writing requires refinement. For instance, while the notation is familiar to experts in the field, the authors should explicitly define its meaning, particularly for and in Equation 4, instead of deferring the explanation to Equation 7.`
>
> We appreciate the reviewer’s feedback and will carefully revise the paper to improve clarity and readability. Specifically, we will ensure that all notation—especially in Equation 4—is clearly defined at the point of introduction, rather than delayed until later sections.
>
> ***
> `Question 1: What is the latency of the proposed method, and how does it perform compared to other methods?`
>
> We report the latency of our model in Table S2 of the supplementary material, including GPU memory usage and total inference time on the PIE-Benchmark.
>
> ***
> `Question 2: What is the fidelity-enhanced setting?`
>
> Thank you for pointing out the lack of clarity regarding the fidelity-enhanced setting. In the baseline method FlowEdit (as described in Lines 125-126 of the main paper), $x_t^{src}$ is obtained by linearly interpolating between the source latent $x_0^{src}$ and a randomly sampled noise. This approximation improves editability in early diffusion steps but may compromise image fidelity to some extent.
>
> Therefore, as mentioned in Line 254 of the main paper, in our fidelity-enhanced setting, we omit this noise interpolation **during the decomposition phase** and directly use the source latent $x_0^{src}$  to better preserve structural details of the input image. For the subsequent unified flow phase, we retain the same settings as the baseline to ensure fairness and comparability of the overall pipeline.
> ***
> `Question 3: What is the performance of applying the proposed method to other base models, such as SD 3.5, Flux, etc.`
>
> We conducted experiments with a stronger base model—Stable Diffusion 3.5. Our method not only outperforms the state-of-the-art method in terms of PSNR, LPIPS, and MSE, but also achieves significantly higher image editability. Also, compared to the baseline (FlowEdit–SD 3.5), SplitFlow delivers meaningful improvements in both fidelity and editability.
>
> | **Method**    | **Model** | **Distance ×10³ ↓** | **PSNR ↑** | **LPIPS ×10³ ↓** | **MSE ×10⁴ ↓** | **SSIM ×10² ↑** | **CLIP Whole ↑** | **CLIP Edited ↑** |
> |---------------|-----------|----------------------|------------|------------------|----------------|------------------|-------------------|--------------------|
> | **FTEdit**    | SD3.5     | 18.17                | 26.62      | 80.55            | 40.24          | 91.50            | 25.74             | 22.27              |
> | **FlowEdit**  | SD3.5     | 9.43                 | 28.31      | 43.59            | 23.40          | 90.84            | 25.88             | 22.50              |
> | **SplitFlow** | SD3.5     | 9.15                 | 28.44      | 43.10            | 22.93          | 90.92            | 25.95             | 22.51              |

---

> > ### Comment · Reviewer_jnDU · 2025-08-05
> > **Reply to rebuttal**
> >
> > I appreciate the authors’ rebuttal and their efforts to address my initial concerns, particularly regarding the theoretical aspect. Nonetheless, several limitations persist.
> >
> > 1. The reliance on the LLM for prompt decomposition significantly constrains the method’s upper performance bound; it is a limitation that remains unaddressed in the current version.
> >
> > 2. Incorporating the theoretical proof would render the paper substantially different, a revision I cannot endorse without reviewing the updated content. Thus, while I am inclined to increase my score, I doubt I will elevate it to an accept recommendation.

---

> > > ### Author Response · Authors · 2025-08-05
> > >
> > > We sincerely appreciate the reviewer’s constructive feedback and your willingness to consider increasing the score.
> > >
> > > We acknowledge that relying on an LLM for prompt decomposition introduces certain limitations. However, we would like to clarify that the core contribution of our work lies in demonstrating that decomposing the editing process into sub-target flows can significantly improve image editing performance—and in proposing a principled method to aggregate these flows effectively. Furthermore, we have already shown that our method remains robust across different LLMs (e.g., LLaMA, Qwen), indicating that the framework is not tightly coupled to a specific decomposition model. We believe this **robustness**, combined with the modularity of our approach, opens up promising directions for future work, where improved decomposition strategies—potentially with more advanced prompting—could further enhance performance.
> > >
> > > Regarding the **theoretical** component, we plan to move Lemma 1 and its proof to the supplementary materials, while including the proof of Eq. (1) and its interpretation in the main paper. To accommodate this revision, we will relocate Fig. 2 to the supplementary section, as suggested by Reviewer JXwp.
> > >
> > > We hope this planned revision demonstrates our commitment to strengthening the theoretical clarity of the paper, while maintaining a coherent presentation.

---

### Decision · Program_Chairs · 2025-09-17

**Decision:**

Accept (poster)

**Comment:**

The paper considers a problem of image editing in rectified-flow diffusion models. To do the edit, authors propose to compute a separate flow for each subprompt and aggregate them into the single editing trajectory via Latent Trajectory Projection (LTP) and Velocity Field Aggregation (VFA). They demonstrate that the approach yields better image editing fidelity than other commonly-used techniques for image editing, such as inversion.

The reviewers raised concerns about the trade-off of the computational overhead and latency to run multiple flows, as well as reliance on LLM-based prompt decomposition.

The authors showed that the model remains efficient in comparison to other editing methods, as well as the need for VFA aggregation over naive averaging. The authors demonstrated that  SplitFlow beats FlowEdit on all metrics on the editing tasks.

The paper is recommended for acceptance.